# Survival Outcomes of U.S. Patients with CMML: A Two-Decade Analysis from the SEER Database

**DOI:** 10.3390/medsci12040060

**Published:** 2024-10-31

**Authors:** Ayrton Bangolo, Behzad Amoozgar, Abhishek Thapa, Wardah Bajwa, Vignesh K. Nagesh, Yaryna Nyzhnyk, Rakshanda Banu, Tirth Bhavsar, Lili Zhang, Olga Velichko, Challa Mani Shankar Reddy, Edwina Essuman, Amal M. Ibrahim, Ramkumar Krishnasamy, Achint Jethi, Arun Ram, Abdullah A. Haq, Abdulla Ahmad Al hashm, Parna Pathak, Shafia Naeem, Rachana R. Gavva, Prajakta H. Ratnaparkhi, Paula Samaha, Cynthia Elizabeth Armendariz Espinoza, Prasansa Dhakal, Frantz Ricot Martine, Mogahid Elkhidir, Jay Mehta, Simcha Weissman

**Affiliations:** 1Department of Hematology and Oncology, John Theurer Cancer Center, Hackensack University Medical Center, Hackensack, NJ 07601, USA; behzad.amoozgar@hmhn.org (B.A.); lili.zhang@hmhn.org (L.Z.); 2Department of Internal Medicine, Hackensack Palisades Medical Center, North Bergen, NJ 07047, USAmdcynthiaarmendariz@gmail.com (C.E.A.E.); simchaweissman@gmail.com (S.W.)

**Keywords:** Chronic Myelomonocytic Leukemia, mortality, prognosis, cytogenetics, incidence, prevalence

## Abstract

**Background:** Chronic Myelomonocytic Leukemia (CMML) is a rare and aggressive form of leukemia with characteristics of both myeloproliferative neoplasms (MPNs) and myelodysplastic syndromes (MDSs). This study aims to explore the clinical features, survival outcomes, and prognostic factors in CMML patients over the past 20 years using a large sample. **Methods:** The study data from 4124 patients diagnosed with CMML between 2000 and 2017 were sourced from the SEER database. Demographic and clinical characteristics, along with overall and cancer-specific mortality, were examined. Factors with a *p*-value < 0.01 in univariate Cox regression were included in the multivariate Cox model to identify independent prognostic factors, with hazard ratios (HRs) greater than one indicating adverse outcomes. **Results:** The majority of the cohort were male (61.57%), and most diagnoses occurred between ages 60–79 (55.16%), with a small percentage under 40 (1.41%). Non-Hispanic whites represented the largest racial group (79.03%). Multivariate analysis showed higher mortality in males, those aged 80+, residents in metropolitan areas with populations between 250,000 and 1 million, single or widowed individuals, and those who underwent chemotherapy. Conversely, lower mortality was associated with an annual income of $75,000+. **Conclusions:** CMML remains a rare and highly aggressive hematologic disorder. This U.S.-based retrospective cohort study identified male gender, advanced age, single or widowed status, and chemotherapy as independent poor prognostic factors. While it is expected that older patients and those requiring chemotherapy would have a poorer prognosis, the higher mortality risk in single or widowed patients, as well as males, warrants further investigation. The early involvement of family and community support may help reduce mortality in these groups, suggesting a need for larger prospective studies to explore these associations further.

## 1. Introduction

CMML is a rare hematological disorder seen predominantly in the elderly, distinguished by increased monocytic cells in the bone marrow and peripheral blood. CMML is a rare leukemia with poorly defined incidence. No known inherited pattern or environmental exposure have been linked to CMML [1,2,3]. However, there have been cases of CMML that arose in the setting of chemotherapy or ionizing radiation [4,5,6].

CMML can be asymptomatic or present with nonspecific symptoms. CMML can have a dysplastic pattern associated with lower White Blood Count (WBC) and consequences of cytopenias such as fatigue, infection, and bleeding [1,2,3]. CMML can also be proliferative with higher WBC levels and constitutional symptoms such as weight loss, fever, and night sweats [7,8,9,10]. Up to half of patients present with Hepatosplenomegaly [1]. A few studies have associated autoimmune diseases with CMML [11,12].

The CMML-specific prognostic scoring system (CPSS) serves as a straightforward yet effective prognostic tool, capable of forecasting both overall survival (OS) and the likelihood of progression to AML. It also categorizes patients into four risk groups with distinctly different outcomes for each measure. This predictive ability was confirmed in an external validation cohort, demonstrating stronger performance compared to other previously published scoring models [8].

Few studies have explored the epidemiology of CMML in depth [13,14]. Nevertheless, there is still a considerable shortage of definitive data and sufficiently large studies that comprehensively outline the epidemiological features, survival outcomes, and prognostic factors of CMML in the last twenty years.

To address this gap in the literature, we utilized a nationally representative, up-to-date database to evaluate the independent prognostic factors in CMML patients. Our goal was to identify patient populations at higher risk for poorer prognosis and provide more definitive insights into the disease’s epidemiology. These patients may require more frequent monitoring and potentially more aggressive treatment, particularly given the advancements in modern therapeutics.

## 2. Materials and Methods

A population-based retrospective cohort study on patients with CMML was carried out using data from the SEER research plus database, covering 18 registries and utilizing the November 2020 submission (http://www.seer.cancer.gov). The SEER Program, sponsored by the U.S. National Cancer Institute (NCI), is one of the largest and most reliable sources for cancer-related data in the United States. The SEER 18 database gathers information on cancer incidence, clinicopathological characteristics, and survival outcomes from 18 population-based cancer registries, representing nearly 28% of the U.S. population [15].

All patients with CMML diagnosed from 2000 to 2017 were selected in our cohort from the SEER database based on histological type [ICD-O-3: 9945). The above-mentioned ICD-0–3 code was used to extract data regarding these patients from the SEER database. We excluded patients whose age or race at diagnosis was unknown, as well as those with cancers other than CMML. Overall Mortality: Patients who had died by this study’s end were categorized as “yes”, while those who survived were categorized as “no”. Cancer-Specific Mortality: Patients who died from CMML were marked as “yes”, while deaths from other causes were marked as “no”. Overall Mortality: Survival time was calculated from diagnosis to death or the last follow-up (31 December 2017), as reported in the SEER registry. Cancer-Specific Mortality: Survival time was calculated from diagnosis to death specifically related to CMML or the last follow-up date in the SEER registry.

Data on variables such as age at diagnosis, gender, race (White, Black, others), ethnicity (Non-Hispanic, Hispanic), geographic location, annual or yearly income, marital status, year of diagnosis, surgery, and radiation therapy were collected.

The Cox proportional hazard regression model assumes that hazard rates remain proportional over time. Variables with a *p*-value less than 0.01 in the univariate Cox regression were included in the multivariate analysis to identify independent prognostic factors for overall mortality (OM) and cancer-specific mortality (CSM). In this analysis, a hazard ratio (HR) greater than 1 signified a negative prognostic factor. All statistical tests were two-sided, with a 95% confidence interval, and a *p*-value of less than 0.05 was considered statistically significant. The analyses were conducted using STATA 18.0 software, developed by StataCorp LLC, based in College Station, TX, USA.

## 3. Results

Our study encompassed 4124 patients with a primary diagnosis of CMML. The baseline characteristics of these patients are detailed in Table 1. The cohort had a male predominance (61.57%), with the majority of patients diagnosed between the ages of 60 and 79 (55.16%). Non-Hispanic whites made up 79.03% of the cohort. Common demographic traits included residence in metropolitan counties with populations over 1 million (54.46%), an annual income of USD 75,000 or more (40.03%), and being married (57.47%). Notably, only four patients underwent cancer-directed surgery.

A preliminary analysis of factors associated with OM and CSM among U.S. patients between 2000 and 2017 is shown in Table 2. The analysis identified significantly higher OM in patients aged 80 and above (HR = 2.68, 95% CI 1.89–3.79, *p* < 0.01), followed by those aged 60–79 (HR = 1.85, 95% CI 1.31–2.62, *p* < 0.01), individuals living in nonmetropolitan areas not adjacent to a metropolitan area (HR = 1.28, 95% CI 1.11–1.48, *p* < 0.01), widowed patients (HR = 1.42, 95% CI 1.30–1.54, *p* < 0.01), and those who received chemotherapy (HR = 1.20, 95% CI 1.13–1.29, *p* < 0.01). Conversely, lower OM was associated with an annual income of USD 75,000+ (HR = 0.85, 95% CI 0.77–0.94, *p* < 0.01) and radiation therapy (HR = 0.64, 95% CI 0.47–0.87, *p* < 0.01). For CSM, higher risks were noted in male patients (HR = 1.12, 95% CI 1.03–1.22, *p* < 0.01), those aged 80 and older (HR = 1.90, 95% CI 1.28–2.81, *p* < 0.01), individuals in nonmetropolitan areas not adjacent to a metropolitan area (HR = 1.30, 95% CI 1.10–1.54, *p* < 0.01), widowed patients (HR = 1.27, 95% CI 1.15–1.41, *p* < 0.01), and those undergoing chemotherapy (HR = 1.59, 95% CI 1.47–1.73, *p* < 0.01).

Table 3 provides a summary of the multivariate Cox proportional hazard regression analyses that examine factors influencing OM and CSM in patients with CMML diagnosed between 2000 and 2017. The analysis shows that OM was significantly higher among males (HR = 1.22, 95% CI 1.13–1.32, *p* < 0.01), patients aged 80 and older (HR = 3.65, 95% CI 2.55–5.21, *p* < 0.01), those living in metropolitan counties with populations between 250,000 and 1 million (HR = 1.15, 95% CI 1.05–1.26, *p* < 0.01), single patients (HR = 1.31, 95% CI 1.16–1.47, *p* < 0.01), widowed patients (HR = 1.28, 95% CI 1.16–1.41, *p* < 0.01), and those who underwent chemotherapy (HR = 1.51, 95% CI 1.39–1.62, *p* < 0.01). In contrast, lower OM was observed in patients with an annual income exceeding USD 75,000 (HR = 0.84, 95% CI 0.73–0.96, *p* < 0.05).

Regarding CSM, increased mortality was seen among male patients (HR = 1.24, 95% CI 1.13–1.35, *p* < 0.05), those aged 80 and older (HR = 3.01, 95% CI 1.99–4.53, *p* < 0.01), individuals residing in metropolitan counties with populations between 250,000 and 1 million (HR = 1.18, 95% CI 1.06–1.32, *p* < 0.01), single patients (HR = 1.30, 95% CI 1.13–1.49, *p* < 0.01), widowed patients (HR = 1.27, 95% CI 1.13–1.43, *p* < 0.01), and those who underwent chemotherapy to treat CMML (HR = 1.91, 95% CI 1.75–2.08, *p* < 0.01).

## 4. Discussion

CMML is a rare leukemia with limited data available on its epidemiology. In this U.S. population-based study, there was a predominance of male and non-Hispanic white patients. This study also found that males, widowed individuals, single patients, and those who underwent chemotherapy experienced higher mortality rates.

CMML primarily affects older adults, with the median age of diagnosis typically falling between 65 and 75 years. The condition is more common in males, with a male-to-female ratio of at least 1.5 [1,2,3]. Higher incidence has been reported in patients 80+ [16]. Our findings mirrored the literature as most patients were diagnosed between 60- and 79-years-old with a second incidence after the age of 80.

Age has been proven to adversely impact the prognosis of CMML patients; a study conducted by Chen et al. revealed a worse prognosis in patients older than 60-years-old [17]. A mutation incriminated in CMML involves mediators of a DNA damage response such as tumor protein p53 (Tp53) [18]. The DEAD-box helicase 41 gene (*DDX41*), located on chromosome 5q3, has been associated with Tp53 mutation and high-risk disease in CMML [19]. Furthermore, DDX41 has been associated with the male gender [19]. This association can explain to a certain degree the poor OM and CSM among male patients in our cohort. Age-related OM associated with TP53 and DDX41 mutations is rather rare or unexplored.

Patients with symptomatic cytopenias often benefit from hypomethylating agents for symptomatic relief [20]. Several prognostic models for CMML recognized cytopenias as an indicator of poor prognosis [21,22]. This observation can explain the higher mortality observed in patients that undergo chemotherapy in our cohort.

Numerous epidemiologic cancer studies identified marital status as an independent factor for reduced mortality, likely due to the enhanced social support experienced by married individuals [23,24,25,26,27,28,29,30,31,32]. Our study similarly found that single and widowed patients had higher cancer-specific mortality (CSM) and overall mortality (OM), with widowed patients showing a particularly elevated OM. These findings highlight the importance of engaging the families of unmarried CMML patients early in the disease process, as social support may play a crucial role in improving survival outcomes.

Our findings suggest that classical scoring systems for CMML could benefit from incorporating broader factors such as socioeconomic status, marital status, and treatment modalities like chemotherapy. While classical models focus heavily on biological markers, this study points to the influence of demographic and treatment-related factors that could refine risk stratification [33,34,35,36]. Integrating these into existing models may provide a more comprehensive and accurate prediction of patient outcomes.

However, some limitations should be noted while interpreting our study results. Information on subjects who received radiotherapy was incomplete, and the publicly available SEER database lacks details on comorbidities or gene mutations. Despite its limitations, the key strength of this study is its utilization of the largest cancer database in the U.S., which ensures an ample sample size, even for a rare condition like CMML.

## 5. Conclusions

The prognosis for CMML is generally poor, with a median survival of less than four years. Our U.S.-based population study identified that advanced age and male gender are associated with higher mortality. While it is expected that older patients would have a higher mortality rate, the increased mortality among male patients warrants further investigation. We hope this study prompts future research to explore how these independent factors of poor prognosis interact and contribute to overall mortality in CMML patients.

## Figures and Tables

**Table 1 medsci-12-00060-t001:** Demographic and clinicopathologic characteristics of US patients diagnosed with CMML between 2000 and 2017.

Characteristics		
	N=	%
Total	4124	100
Gender		
Female	1585	38.43
Male	2539	61.57
Age at diagnosis, y.o		
0–39	58	1.41
40–59	372	9.02
60–79	2275	55.16
80+	1419	34.41
Race		
Non-Hispanic white	3259	79.03
Non-Hispanic black	242	5.87
Hispanic	340	8.24
Other	283	6.86
Living area		
Counties in metropolitan areas of 1 million persons	2246	54.46
Counties in metropolitan areas of 250,000 to 1 million persons	872	21.14
Counties in metropolitan areas of 250,000 persons	393	9.53
Nonmetropolitan counties adjacent to a metropolitan area	359	8.71
Nonmetropolitan counties not adjacent to a metropolitan area	254	6.16
Income per year		
USD <55,000	638	15.47
USD 55,000–64,999	693	16.80
USD 65,000–74,999	1142	27.69
USD 75,000+	1651	40.03
Marital status		
Married	2370	57.47
Single	445	10.79
Divorced/separated	364	8.83
Widowed	945	22.91
Radiation		
No	4065	98.57
Yes	59	1.43
Chemotherapy		
No	2537	61.52
Yes	1587	38.48
Surgery +/− radiation		
No	4122	99.95
Yes	2	0.05
Surgery		
No	4120	99.90
Yes	4	0.10
Year of diagnosis		
2000	175	4.24
2001	192	4.66
2002	185	4.49
2003	205	4.97
2004	198	4.80
2005	205	4.97
2006	204	4.95
2007	218	5.29
2008	222	5.38
2009	251	6.09
2010	224	5.43
2011	238	5.77
2012	212	5.14
2013	250	6.06
2014	265	6.43
2015	271	6.57
2016	317	7.69
2017	292	7.08

**Table 2 medsci-12-00060-t002:** Crude analysis of factors associated with all-cause mortality and CMML-related mortality among US patients between 2000 and 2017.

Characteristics	Overall Mortality. Crude Proportional Hazard Ratio (95% Confidence Interval)	CMML Mortality. Crude Proportional Hazard Ratio (95% Confidence Interval)
Gender		
Female	1	1
Male	1.07 (0.99–1.15)	1.12 (1.03–1.22) *
Age at diagnosis, y.o		
0–39	1	1
40–59	1.07 (0.75–1.55)	1.11 (0.74–1.68)
60–79	1.85 (1.31–2.62) **	1.59 (1.08–2.34) *
80+	2.68 (1.89–3.79) **	1.90 (1.28–2.81) **
Race		
Non-Hispanic white	1	1
Non-Hispanic black	1.03 (0.89–1.19)	1.04 (0.87–1.23)
Hispanic	0.94 (0.83–1.07)	1.03 (0.89–1.19)
Other	0.98 (0.86–1.13)	0.99 (0.84–1.17)
Living area		
Counties in metropolitan areas of 1 million persons	1	1
Counties in metropolitan areas of 250,000 to 1 million persons	1.19 (1.09–1.29) **	1.21 (1.09–1.35) **
Counties in metropolitan areas of 250,000 persons	1.05 (0.94–1.19)	1.03 (0.89–1.19)
Nonmetropolitan counties adjacent to a metropolitan area	1.13 (0.99–1.27)	1.12 (0.96–1.30)
Nonmetropolitan counties not adjacent to a metropolitan area	1.28 (1.11–1.48) **	1.30 (1.10–1.54) **
Income per year		
USD <55,000	1	1
USD 55,000–64,999	0.88 (0.78–0.99) *	0.93 (0.80–1.07)
USD 65,000–74,999	0.89 (0.81–0.99)	0.91 (0.79–1.04)
USD 75,000+	0.85 (0.77–0.94) **	0.89 (0.79–1.01)
Marital status		
Married	1	1
Single	1.11 (0.99–1.25)	1.14 (0.99–1.31)
Divorced/Separated	1.02 (0.90–1.16)	0.99 (0.85–1.15)
Widowed	1.42 (1.30–1.54) **	1.27 (1.15–1.41) **
Chemotherapy		
No	1	1
Yes	1.20 (1.13–1.29) **	1.59 (1.47–1.73) **
Radiation		
No	1	1
Yes	0.64 (0.47–0.87) **	0.80 (0.57–1.12)

** *p* < 0.01, * *p* < 0.05.

**Table 3 medsci-12-00060-t003:** Multivariate cox proportional hazard regression analyses of factors affecting all-cause mortality and CMML-related mortality among US patients between 2000 and 2017.

Characteristics	Overall Mortality. Adjusted Proportional Hazard Ratio (95% Confidence Interval)	CMML Mortality. Adjusted Proportional Hazard Ratio (95% Confidence Interval)
Gender		
Female	1	1
Male	1.22 (1.13–1.32) **	1.24 (1.13–1.35) **
Age at diagnosis, y.o		
0–39	1	1
40–59	1.26 (0.87–1.82)	1.37 (0.90–2.07)
60–79	2.39 (1.68–3.41) **	2.29 (1.53–3.41) **
80+	3.65 (2.55–5.24) **	3.01 (1.99–4.53) **
Race		
Non-Hispanic white	1	1
Non-Hispanic black	1.13 (0.97–1.32)	1.13 (0.94–1.36)
Hispanic	1.03 (0.89–1.17)	1.08 (0.93–1.26)
Other	1.05 (0.92–1.21)	1.06 (0.89–1.25)
Living area		
Counties in metropolitan areas of 1 million persons	1	1
Counties in metropolitan areas of 250,000 to 1 million persons	1.15 (1.05–1.26) **	1.18 (1.06–1.32) **
Counties in metropolitan areas of 250,000 persons	0.99 (0.86–1.13)	1.01 (0.86–1.19)
Nonmetropolitan counties adjacent to a metropolitan area	0.99 (0.86–1.16)	1.01 (0.84–1.21)
Nonmetropolitan counties not adjacent to a metropolitan area	1.13 (0.95–1.34)	1.19 (0.97–1.46)
Income per year		
USD <55,000	1	1
USD 55,000–64,999	0.86 (0.76–0.98) *	0.93 (0.79–1.08)
USD 65,000–74,999	0.89 (.077–1.02)	0.93 (0.79–1.09)
USD 75,000+	0.84 (0.73–0.96) *	0.91 (0.77–1.07)
Marital status		
Married	1	1
Single	1.31 (1.16–1.47) **	1.30 (1.13–1.49) **
Divorced/Separated	1.11 (0.98–1.27)	1.06 (0.91–1.24)
Widowed	1.28 (1.16–1.41) **	1.27 (1.13–1.43) **
Chemotherapy for CMML		
No	1	1
Yes	1.51 (1.39–1.62) **	1.91 (1.75–2.08) **
Radiation		
No	1	1
Yes	0.89 (0.65–1.23)	0.93 (0.65–1.31)

** *p* < 0.01, * *p* < 0.05.

## Data Availability

The data used and/or analyzed in this study are available in the Surveillance, Epidemiology, and End Results (SEER) Database of the National Cancer Institute (http://seer.cancer.gov).

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
