# Peer review of "Survival Outcomes of U.S. Patients with CMML: A Two-Decade Analysis from the SEER Database"

_medsci, 2024, doi:10.3390/medsci12040060_

Round 1

Reviewer 1 Report

Comments and Suggestions for Authors

Bangolo, et al report a impacts of multiple parameters of survival outcomes of CMML patients in US, usnig big database. Some points need to clarified

1.      Please make a brief define of CMML.

2.      Line 68, please write the ICD-9 and ICD-10 code that you enrolled.

3.      Line 79-80, do you use primary GI melanoma as caner-specific mortaligy? Do you miss-type?

4.      Line 98, please mention and company and nation of the software (STATA 19.0) you used.

5.      Did your patients have 2nd cancers? What is the chemotherapy regimen/drugs? was that used to treat CMML? If yes, did the patients who received chemotherapy show acute leukemia change? How long did they receive chemotherapy? Why do CMML patients use radiotherapy? Is that used for splenomegaly or something else? Did you count radiotherpay after CMML diagnosed as “yes”? Or you defined “received R/T” if patients ever received R/T, no matter when?

6.      Line 109, please write the full name when you use abbreation (OM, CSM)

7.      Line 153-159, genes alternation can not explain why older than 60 year-old population had poor outcome in your cohort.

8.      Lin 163-164, do you hint that the chemotherapy in your cohort were hypomethylating agents?

Author Response

  1. Please make a brief define of CMML.

Re: Thank you for this keen observation, a definition of CMML was added to the introduction and highlighted in red.

  1. Line 68, please write the ICD-9 and ICD-10 code that you enrolled.

Re: Thank you for this keen observation, ICD-03 codes accounting for the histology and behavior of the tumor enrolled in our study have been added and highlighted in red and this is the classification used by SEER database utilized in our study. The SEER database does not offer ICD-9 or ICD-10 codes and was an error from our end.

  1. Line 79-80, do you use primary GI melanoma as cancer-specific mortality? Do you miss-type?

Re: Thank you for this keen observation , that was an error , has been corrected  to CMML.

  1. Line 98, please mention and company and nation of the software (STATA 19.0) you used.

Re: The STATA 18 software used is developed by Statcorp in USA and has been added to the methods section and highlighted in red

  1. Did your patients have 2nd cancers? What is the chemotherapy regimen/drugs? was that used to treat CMML? If yes, did the patients who received chemotherapy show acute leukemia change? How long did they receive chemotherapy? Why do CMML patients use radiotherapy? Is that used for splenomegaly or something else? Did you count radiotherapy after CMML diagnosed as “yes”? Or you defined “received R/T” if patients ever received R/T, no matter when?

    Thank you for the valid points , however the SEER database does not provide data on the type of chemotherapeutic regimens or the time period they received the chemotherapy. Radiation is used in CMML for symptomatic relief of splenomegaly and as part of conditioning regimen before bone marrow transplant. There were no secondary malignancies in our patient cohort,
  2. Line 109, please write the full name when you use abbreviation (OM, CSM)

Re: Thank you for this keen observation, changes have been made and highlighted in red

  1. Line 153-159, genes alternation can not explain why older than 60 year-old population had poor outcome in your cohort.

Re: Thank you for this observation, the gene alternation could explain the higher mortality in males which happens over time as mentioned in our discussion section. Age has been associated with higher mortality in the literature as mentioned in the discussion section

  1. Lin 163-164, do you hint that the chemotherapy in your cohort were hypomethylating agents?

Re: The SEER database does not provide data on the chemotherapy regimen used but hypomethylating agents are used in the treatment of CMML in the USA.

Reviewer 2 Report

Comments and Suggestions for Authors

The work of Bangolo et al. is an epidemiologic study of 4124 CMML patients with data obtained from the SEER database. The authors identify convincingly independent risk factors compared to classical risk scores (GFM score, CMML-CPSS, etc.). Age, male gender, living in larger metropolitan areas and received chemotherapy (prior of or for CMML?) are associated with increased overall mortality (OM) while higher income was associated with lower OM. The study is interesting as epidemiologic data for CMML are sparse. Nevertheless results should be investigated and compared with other regions in the future to enhance impact. Several points should be addressed:

1.     In the introduction authors should describe risk factors (GFM score, CMML-CPSS, etc.)

2.     Pp5 ll112-113 “individuals living in nonmetropolitan areas (HR = 1.28, 95% CI 1.11-1.48, p < 0.01)” and ll 118-119 “individuals in non-metropolitan areas (HR = 1.30, 95% CI 1.10-1.54, p < 0.01)” this seems conflicting and it should clearly be stated that OM is increased in some metropolitan areas 

3.     Table 3 “chemotherapy” it should be clarified in the text and table what means chemotherapy, is it chemotherapy for another cancer and CMML is therapy related or does it mean chemotherapy for CMML (or both? Then chemotherapy groups should be analyzed separately)

4.     “A common mutation incriminated in CMML involves mediators of DNA damage response such as tumor protein p53 (Tp53) [18]. The DEAD-box helicase 41 gene (DDX41), located on chromosome 5q3 has been associated with Tp53 mutation and high-risk disease in CMML [19]. Furthermore, DDX41 has been associated with the male gender [19]. This can explain to a certain degree the poor OM and CSM among male patients in our cohort.” This paragraph should be tempered, neither TP53 mutations and DDX41 mutations are common in CMML thus age-related OM associated with these mutations is rather rare or unexplored

5.     Are gene mutations in CMML reported in the SEER data base? Could gene mutations be included in the analysis?

6.     Discussion should be extended to the comparison of the author’s findings and potential use with classical scoring systems

Author Response

The work of Bangolo et al. is an epidemiologic study of 4124 CMML patients with data obtained from the SEER database. The authors identify convincingly independent risk factors compared to classical risk scores (GFM score, CMML-CPSS, etc.). Age, male gender, living in larger metropolitan areas and received chemotherapy (prior of or for CMML?) are associated with increased overall mortality (OM) while higher income was associated with lower OM. The study is interesting as epidemiologic data for CMML are sparse. Nevertheless results should be investigated and compared with other regions in the future to enhance impact. Several points should be addressed:

Re: Thank you very much for recognizing and acknowledging the relevance of our study.

  1. In the introduction authors should describe risk factors (GFM score, CMML-CPSS, etc.)

Re: Thank you for this keen suggestion, one line was added and highlighted in yellow.

  1. Pp5 ll112-113 “individuals living in nonmetropolitan areas (HR = 1.28, 95% CI 1.11-1.48, p < 0.01)” and ll 118-119 “individuals in non-metropolitan areas (HR = 1.30, 95% CI 1.10-1.54, p < 0.01)” this seems conflicting and it should clearly be stated that OM is increased in some metropolitan areas.

Re: Higher OM and CSM were observed in nonmetropolitan areas not adjacent to a metropolitan area. And thin was clarified in the manuscript and highlighted in yellow.

  1. Table 3 “chemotherapy” it should be clarified in the text and table what means chemotherapy, is it chemotherapy for another cancer and CMML is therapy related or does it mean chemotherapy for CMML (or both? Then chemotherapy groups should be analyzed separately)

Re: It means CMML chemotherapy, patients with other types of cancer were not included in our study. This was clarified in the table and the text.

  1. “A common mutation incriminated in CMML involves mediators of DNA damage response such as tumor protein p53 (Tp53) [18]. The DEAD-box helicase 41 gene (DDX41), located on chromosome 5q3 has been associated with Tp53 mutation and high-risk disease in CMML [19]. Furthermore, DDX41 has been associated with the male gender [19]. This can explain to a certain degree the poor OM and CSM among male patients in our cohort.” This paragraph should be tempered, neither TP53 mutations and DDX41 mutations are common in CMML thus age-related OM associated with these mutations is rather rare or unexplored

Re: The word “common” was removed. The following sentence “Age-related OM associated with TP53 and DDX41 mutations is rather rare or unexplored” was added.

  1. Are gene mutations in CMML reported in the SEER data base? Could gene mutations be included in the analysis?

Re: Unfortunately, gene mutations are not reported in the SEER database and this was added as a weakness of our study.

  1. Discussion should be extended to the comparison of the author’s findings and potential use with classical scoring systems

Re: Thank you for this keen suggestion, a paragraph was added to enhance the quality of our paper.

Reviewer 3 Report

Comments and Suggestions for Authors

This paper presents a retrospective cohort study of CMML (chronic myelomonocytic leukemia) patients over the past 20 years using data from the SEER database. The authors conducted both crude and multivariate analyses to investigate factors affecting overall and cancer-related mortality. They identified sex, age, living area, and marital status as independent predictors of survival, while income and race were not associated with cancer-related mortality. Notably, patients who received chemotherapy had poorer prognosis outcomes. The study's large sample size strengthens its findings, which are consistent with previous research, highlighting age, sex, and chemotherapy as significant risk factors for increased cancer-related mortality. The study provides valuable insights into factors associated with mortality in CMML patients.

Minor Comments:

1, Line 20: Could the authors confirm whether factors with a p-value < 0.1 or p-value < 0.01 in the univariate Cox regression were included in the multivariate analysis? Line 93 indicates that a p-value < 0.1 was used for subsequent analyses. Clarification on this process would be helpful.

2, Lines 77 and 79: The abbreviations for Overall Mortality (OM) and Cancer-Specific Mortality (CSM) appear on line 109. Could the authors provide the full forms of these abbreviations where they first appear in the manuscript for clarity?

3, Could the authors consider performing multiple logistic regression analysis to explore the association between risk factors and mortality, as expressed by adjusted odds ratios?

Author Response

1, Line 20: Could the authors confirm whether factors with a p-value < 0.1 or p-value < 0.01 in the univariate Cox regression were included in the multivariate analysis? Line 93 indicates that a p-value < 0.1 was used for subsequent analyses. Clarification on this process would be helpful.

Re: Thank you for this keen observation, factors with a p-value <0.01 were incorporated into multivariate analysis. Changes have been made and highlighted in turquoise.

    2, Lines 77 and 79: The abbreviations for Overall Mortality (OM) and Cancer-Specific Mortality (CSM) appear on line 109. Could the authors provide the full forms of these abbreviations where they first appear in the manuscript for clarity?
Re: Thank you for this keen observation, the abbreviations have been expanded and highlighted in red.

  3, Could the authors consider performing multiple logistic regression analysis to explore the association between risk factors and mortality, as expressed by adjusted odds ratios?

Re:  Multivariate logistic regression was performed in our study. However, the association between risk factors and mortality was beyond the scope of our study and can be considered in subsequent studies in the future.

Round 2

Reviewer 1 Report

Comments and Suggestions for Authors

Dear Authors

Thank you for your reply. However, few points need to be clarified.

As you mention, R/T is for symptom relief and conditioning regimen before bone marrow transplantation. We know that bone marrow transplantation or the condition need transplantation is a risk factor and related to survival outcome. Is that possible to find data in your data base? Including bone marrow transplantation to analysis will be more delicate of your manuscript.

Author Response

Thank you for your reply. However, few points need to be clarified.

As you mention, R/T is for symptom relief and conditioning regimen before bone marrow transplantation. We know that bone marrow transplantation or the condition need transplantation is a risk factor and related to survival outcome. Is that possible to find data in your data base? Including bone marrow transplantation to analysis will be more delicate of your manuscript.

Re: Thank you for this keen point. However, the SEER database does not provide information on bone marrow transplant. 

Reviewer 2 Report

Comments and Suggestions for Authors

The authors addressed all issues raised.

Author Response

The authors addressed all issues raised.

Re: Thank you very much for this observation.